# Toward Flexible and Efficient Home Context Sensing: Capability Evaluation and Verification of Image-Based Cognitive APIs [note 1]

**DOI:** 10.3390/s20051442

**Published:** 2020-03-06

**Authors:** Sinan Chen, Sachio Saiki, Masahide Nakamura

**Affiliations:** 1Graduate School of System Informatics, Kobe University, 1-1 Rokkodai-cho, Nada, Kobe 657-8501, Japan; sachio@carp.kobe-u.ac.jp (S.S.); masa-n@cs.kobe-u.ac.jp (M.N.); 2RIKEN Center for Advanced Intelligence Project, 1-4-1 Nihonbashi, Chuo-ku, Tokyo 103-0027, Japan

**Keywords:** smart home, contexts, cognitive API, image, document similarity, internal cohesion, external isolation, clustering

## Abstract

Cognitive Application Program Interface (API) is an API of emerging artificial intelligence (AI)-based cloud services, which extracts various contextual information from non-numerical multimedia data including image and audio. Our interest is to apply image-based cognitive APIs to implement flexible and efficient context sensing services in a smart home. In the existing approach with machine learning by us, with the complexity of recognition object and the number of the defined contexts increases by users, it still requires directly manually labeling a moderate scale of data for training and continually try to calling multiple cognitive APIs for feature extraction. In this paper, we propose a novel method that uses a small scale of labeled data to evaluate the capability of cognitive APIs in advance, before training features of the APIs with machine learning, for the flexible and efficient home context sensing. In the proposed method, we exploit document similarity measures and the concepts (i.e., internal cohesion and external isolation) integrate into clustering results, to see how the capability of different cognitive APIs for recognizing each context. By selecting the cognitive APIs that relatively adapt to the defined contexts and data based on the evaluation results, we have achieved the flexible integration and efficient process of cognitive APIs for home context sensing.

## 1. Introduction

With the rapid progress of ICT and Internet of Things (IoT) technologies, research and development of smart homes have been actively conducted. In smart homes, it is common to use ambient and/or wearable sensors such as temperature, humidity, motion, and accelerometer in order to retrieve contexts of users and homes for achieving various value-added services such as References [1,2,3]. In recent years, Artificial Intelligence (AI) and cloud computing technologies have brought enormous development potentiality for smart home services. With the progress of emerging deep learning, retrieving the informative features of home contexts is not limited to conventional sensors data [4,5], but includes multimedia data such as the research in References [6,7,8]. Using multimedia data, such as image and audio, for home context sensing is promising for value-added smart services, since the multimedia data contain richer information than the conventional sensor data. However, recognizing multimedia data generally requires massive computation. It was thus unrealistic for general households to install and maintain such a complex and tedious system at home. In recent years, a cognitive service provides the capability to understand multimedia data based on sophisticated machine-learning algorithms powered by big data and large-scale computing resources. Typical services include image recognition [9], speech recognition [10], and natural language processing [11]. A cognitive service usually provides cognitive APIs (Application Program Interface), with which developers can easily integrate powerful recognition features in their own applications. We consider that cognitive APIs make full use of multimedia data. Therefore, they have great potential to improve smart homes since the user would no longer need to maintain a complex and tedious system. Although various kinds of cognitive APIs exist and have been researching such as [12,13,14], we especially focus on image recognition APIs in this paper. An image recognition API receives an image from an external application, extracts specific information from the image, and returns the information as a set of words called tags. The information of interest varies between services. For example, Microsoft Azure Face API [15] estimates age, sex, and emotional values from a given human face image. IBM Watson Visual Recognition [16] recognize items in the image such as home appliances, furniture, and tools.

The main contribution of this paper is to propose a novel method that uses a small scale of labeled data, to evaluate the capability of cognitive APIs in advance, before training features of the APIs with machine learning, for the flexible and efficient home context sensing. We generally divide the *t* (the total number of labeled data) into three levels: (1) large scale (t> 10,000), (2) moderate scale (10,000 ≥t≥1000), (3) small scale (t≤100). In the existing home context sensing approach with machine learning [17,18], with the complexity of recognition object and the number of the defined contexts increase by users, it still requires directly manually labeling a moderate scale of data for training and continually try to calling multiple cognitive APIs for feature extraction. However, for each defined home context, according to different capabilities by cognitive APIs, there will be a difference among difficult-to-train data. That is, the individual contexts with low recognition accuracy by the different APIs, which requires us to make a capability evaluation of each cognitive API and defined context in advance before manually labeling a lot of data. Using the proposed method, one can understand the coverage and limitation of different APIs towards specific home contexts for flexible integration. Also, one can reduce unnecessary data manual labeling and calling cognitive APIs for an efficient process.

The previous version of this paper was published as a conference paper [19]. Changes made to this version are most significantly the addition of clustering algorithms and the model construction based on the former. In the proposed method, we first capture images of different contexts. Afterward, we send the image to the cognitive API to retrieve tags from the images. Finally, integrating the internal cohesion and external isolation concepts into the clustering results, we evaluate the capability of the APIs by checking if the tags can sufficiently characterize(or distinguish) the context shown in the original image. Our key idea of evaluation is to integrate document similarity measures [20,21,22], the concepts (i.e., internal cohesion and external isolation [23,24,25,26]) into results of clustering [27,28], to see how the capability of different cognitive APIs for recognizing each context. More specifically, we evaluate the clustering algorithm results, with respect to the internal cohesion and external isolation. That is, we see if cluster belonging to the same (or different) context(s) are associated with similar tags (or dissimilar tags, respectively). Based on the evaluation results, we produced a flexible and efficient way to select the high capability APIs for building a high accuracy model with machine learning.

Based on the proposed method, we have experimented with the smart home space of our laboratory. Follow the proposed steps, we have completed the capability evaluation and verification of cognitive APIs. The experimental results showed that the five of seven contexts recognition accuracy reached 100%, the remaining difficult contexts also well within the response range of evaluated results. This fully shows the reliability of our proposed method. The remainder of this paper is organized as follows. Section 2 introduces the related work of cognitive APIs from recent years. Section 3 produces a complete description of the proposed method. The experimental evaluation and verification of image-based cognitive APIs for home context sensing are presented in Section 4, followed by conclusions in Section 5.

## 2. Related Work

To retrieve contexts of users and homes for achieving various value-added services, simplify the process and improve the quality of the original is significant. As described in the introduction, with the kind and number of the home context increase from one house to another, it is always a key difficult issue to simplify the model and achieve a more efficient process. In this section, we introduce some related works in the smart home field from recent years around the above issues.

Tax et al. [29] provide a novel algorithm to extract those relevant parts of the data for support counting, only consider specific parts of the datasets instead of the full dataset, which allows speed up the counting of the support of a pattern. Unlike their approach, we use all dataset made by labeling selected by manually that applies into machine learning algorithms. The core of this paper is to present a method that by evaluating, in order to help the user to know how to select original data better in advance. The research in Reference [30] proposes a distributed service-oriented architecture (D-SOA) for a smart-home system, which improved communication efficiency, reduction in network load, and response time. Comparing with their approach, we got the same achievements, by reducing the process to call the low-capability APIs for specific contexts. Xu et al. [31] proposed the software-defined smart home platform, which flexibly adapts to the great difference between family scenes and user demands. We are also focusing on this key point. Unlike their research, we use a fixed-point camera rather than smart devices with an interface for our study, in order to reduce the complexity of system operation by the user. However, for the implementation, the number and setting position of the camera is a big issue. The research in Reference [32] concerned with the use of traffic classification techniques for inferring events taking place within a building, in order to improve security and privacy concerns of the smart homes. In our study, the security and privacy all depend on each cloud service environment of cognitive APIs. Due to the local system does not save any images and information of users, to reduce the unnecessary APIs calling can significantly improve this issue by the proposed method in this paper. Stojkoska et al. [33] present a three-tier Internet of Thing based hierarchical framework for the smart home using fog computing. Although local computation is cheaper operation than communication, it is difficult to retrieve rich and full feature values for fine-grained home context sensing in a short time. It might make simple problems become complicated. To simplify the process of using cloud computing, and improve the quality of the labeled data is the core of this paper.

## 3. Methodology

This section describes the previous method of this paper [19], emphatically presents the proposed method in this paper, and discusses the related techniques.

### 3.1. Previous Method

Since the existing APIs are trained for general-purpose image recognition, they may not be of practical use in the specific configuration of smart homes. In the previous version of this paper, we presented a method that evaluates and compares the capability of multiple image recognition APIs using a few image data, for a given set of home contexts. Figure 1 depicts the essential part of the previous method. In the figure, $\left\{c_{1},c_{2},\cdots ,c_{m}\right\}$ represent a given set of home contexts. For each context, we collect *n* images at home, then send the images to cognitive APIs. Finally, we evaluate the performance of the APIs, by analyzing the output tags. More specifically, the previous method consists of the following five steps:

**Step** **1:**
**Acquiring images**


A user of the proposed method deploys an image capturing device (e.g., USB camera) in the target space, and configures the device to take snapshots of the space periodically with an appropriate interval.

**Step** **2:**
**Defining home contexts to recognize**


The user defines a set $C=\{c_{1},c_{2},\ldots ,c_{m}\}$ of home contexts to be recognized by the cognitive API, such as “Dining”, “Cleaning”, “Nobody” and so on.

**Step** **3:**
**Selecting representative images**


For each context $c_{i}\in C$, the user manually selects representative *n* images $IMG\left(c_{i}\right)=\left\{img_{i1},img_{i2},\ldots ,img_{in}\right\}$ that well expose $c_{i}$, from all images obtained in Step 1. Note, the user needs to select images on different days as possible, to avoid overfitting the data. To evaluate the capability of cognitive APIs, the user can first select a small scale of image data (Generally, n≤10 for keeping m×n with a small scale range). The specific number depends on the situation.

**Step** **4:**
**Calling cognitive API**


The user designates a set $API=\{api_{1},api_{2},\cdots ,api_{q}\}$ of cognitive APIs to be evaluated. There are many cognitive APIs that extract tags from images. For every $c_{i}\in C$, $img_{ij}\in IMG\left(c_{i}\right)$, and $api_{k}\in API$, $api_{k}\left(img_{ij}\right)$ is invoked, and a set $Tag\left(img_{ij},api_{k}\right)=\left\{w_{1},w_{2},w_{3},\ldots \right\}$ of output tags is obtained. $Tag(img_{ij},api_{k})$ represents a recognition result for cognitive API $api_{k}$ for an image $img_{ij}$ belonging to a context $c_{i}$. The size of $Tag(img_{ij},api_{k})$ varies for $img_{ij}$ and $api_{k}$. Since there are *m* contexts, *n* images for each context, and *q* APIs, this step creates totally m×n×q sets of output tags.

**Step** **5:**
**Analyzing output tags**

**Step 5-1: Encoding output tags**
The vector is a numerical representation of the document, where each component of the vector refers to a tag. It shows the presence or importance of that tag in the document. More specifically, regarding every set $Tag(img_{ij},api_{k})$ of output tags as a document corpus, the user can extract features from each document, by converting every set $Tag(img_{ij},api_{k})$ into a document vector $V\left(img_{ij},api_{k}\right)=\left\{v_{1},v_{2},\ldots \right\}$, using a document vectorizing technique, such as *TF-IDF* [34], *Word2Vec* [35], *Doc2Vec* [36], *GloVe* [37], *fastText* [38] and so on. Table 2 shows an example of encoding output tags by the TF-IDF method with python.
**Step 5-2: Document similarity measure**
Regarding each document vector in ${⋃}_{ij}V\left(img_{ij}\right)$ of each $api_{k}$, the method calculates the similarity or distance, which is denoted as ‘≈’, between any two of documents using a certain method of the document similarity measure. Regarding the calculation of *document similarity*, there exists a variety of methods in the field of natural language processing, such as *Cosine Similarity* [39], *Euclidean Distance* [40], *Pearson Correlation Coefficient* [41] and so on.
**Step 5-3: Analyzing document similarity**
For each $api_{k}$, the user evaluates the performance of $api_{k}$ of context recognition, with respect to internal cohesion and external isolation. The internal cohesion represents a capability that $api_{k}$ can produce similar output tags for images in the same context. That is, for $c_{i}\in C$, we evaluate $Tag(img_{ij},api_{k})$≈$Tag(img_{ij^{\prime }},api_{k})$. On the other hand, the external isolation represents a capability that $api_{k}$ can produce dissimilar output tags for images in different contexts. That is, for $c_{x}\neq c_{y}$, we evaluate $Tag(img_{xj},api_{k})$¬≈$Tag(img_{yj^{\prime }},api_{k})$.

**Listing 1 sensors-20-01442-t002:** Example of encoding output tags by the TF-IDF method with python

**from** sklearn.feature_extraction.text **import** TfidfVectorizer
**import** numpy as np
**import** pandas as pd

**for** api **in** ["API_name"]:
tags = np.array(tags_labels_pd[api])
contexts = np.array(tags_labels_pd["labels"])
vectorizer = TfidfVectorizer(use_idf=True)
vecs_tfidf = vectorizer.fit_transform(tags)
np.set_printoptions(precision=3)
np.set_printoptions(threshold=np.inf)
tfidf_vectors = vecs_tfidf.toarray()
feature_names = vectorizer.get_feature_names()
feature_names = **list**(feature_names)
feature_names.append("labels")

vectors_pd = pd.DataFrame(tfidf_vectors)
vectors_pd = vectors_pd.**round**(3)

labels_pd = pd.DataFrame(contexts)
vectors_labels_pd = pd.concat([vectors_pd,labels_pd],axis=1)

vectors_labels_pd.columns = [feature_names]

vectors_labels_pd

### 3.2. Proposed Method

As follow-up studies [17,18] continue, our work is not limited to simply evaluating the capability of cognitive APIs, but more focus on the implementation of a flexible and efficient process for home context sensing. As we mentioned in Section 1, we are struggling to understand the coverage and limitation of different APIs towards specific home contexts, and reduce unnecessary data manual labeling and calling cognitive APIs process. Based on Step 1 to Step 5-1 in the previous method, changes made to this version are most significantly the addition of the new Step 5-2 and 5-3, for evaluating the capability of cognitive APIs, and presents the Step 6-1 to Step 6-4 for building model to verifying the evaluation results. The following explains the proposed new steps in detail.

**Step** **5:**
**Analyzing output tags**

**New Step 5-2: Clustering document vectors**
The user first randomizes the order of all document vectors ${⋃}_{ij}V\left(img_{ij}\right)$ of each $api_{k}$ along with the corresponding labels, then split them into the non-labeled document vectors and the known labels. After that, the user applies a clustering algorithm *W* into the randomized non-labeled document vectors of each $api_{k}$. The algorithm *W* include *k-means* [27], *Partitioning Around Medoids (PAM)* [42], *Clustering Large Applications (CLARA)* [43] and so on. This step is a process of automatic classification, which requires the user to define the number of clusters to classify in advance. By using the clustering algorithm, it returns the integer labels corresponding to the different clusters in the results. Table 3 shows an example of applying k-means++ algorithm into document vectors with python. Further more, an example of the clustering results in cross-tab is shown in Figure 2a, which produced by integrating the known labels and the returned integer labels. The evaluation key is the number of all labels more concentrated in the different classes of both rows and columns, the better the classification effect. Since the naive clustering algorithm cannot show the classification effect well to a small scale of data in general, the new Step 5-3 improves it.
sensors-20-01442-t003_Listing 2Listing 2Example of applying k-means++ algorithm into document vectors with python**from** sklearn.metrics.pairwise **import** cosine_similarity**from** sklearn.cluster **import** kmeans**import** numpy as np**import** pandas as pd random_vectors_labels_pd = vectors_labels_pd.sample(frac=1)random_vectors_np = random_vectors_labels_pd.iloc[:,:-1].valuesrandom_labels_np = random_vectors_labels_pd.iloc[:,-1:].values kmeans.euclidean_distances = cosine_similaritymodel = kmeans(n_clusters=**len**(**sorted**(**list**(**set**(random_labels_np)))),init=’k-means++’)model_output_labels_np =model.fit_predict(non_labels_random_vectors_pd) model_output_labels_pd = pd.DataFrame(model_output_labels_np,columns=[’Assignments’])random_labels_pd = pd.DataFrame(random_labels_np, columns=[’labels’]) labels_concat_pd = pd.concat([random_labels_pd,model_output_labels_pd], axis=1)result_crosstab_pd = pd.crosstab(labels_concat_pd[’assignments’],labels_concat_pd[’labels’]) result_crosstab_pd
**New Step 5-3: Analyzing clustering results**
From the results of automatic classification of each $api_{k}$, the user evaluates the recognition capability of $c_{i}\left(c_{i}\in C\right)$. The core of the evaluation method is to integrate the internal cohesion and external isolation concepts into the clustering results. Specifically, Table 4 shows an example of the key method for evaluating the capability of cognitive APIs with python. Further more, an example of the process for evaluating the capability of cognitive APIs shown in Figure 2, including a calculation formula and the principles. As the evaluation description of this step, refer to the cross-tab of Figure 2b, the capability evaluation method includes two points:(1) The maximum value in each row shows the capability of $api_{k}$ for the row $c_{i}$. Note, it cannot conduct the evaluation using this score if no maximum value in that row.(2) The more there are other values in the row or column of the maximum value of each row, the lower the capability of $api_{k}$ for $c_{i}$ of the row where the maximum value is.Here, $c_{d}\left(c_{d}\in C\right)$ with low scores may be regarded as difficult-to-train contexts.

**Listing 3 sensors-20-01442-t004:** Example of the key method for evaluating the capability of cognitive APIs with python

**import** numpy as np
**import** pandas as pd
**import** seaborn as sns

sum_row_values = result_crosstab_pd.**sum**(axis=1)
sum_column_values = result_crosstab_pd.**sum**(axis=0)

final_results_list = []
**for** i **in** **range**(**len**(result_crosstab_pd)):
this_row_results = []
this_row = result_crosstab_pd.iloc[i:i+1,:]
sum_this_row_values = **float**(sum_row_values[i])
**for** j **in** **range**(**len**(result_crosstab_pd)):
this_value = this_row.iloc[:,j:j+1]
this_value = **float**(this_value.values)
sum_this_column_values = **float**(sum_column_values[j])
**if** this_value != 0:
this_value = ((this_value/sum_this_row_values)
*(this_value/sum_this_column_values))
**else:pass**
this_row_results.append(**round**(this_value,1))
final_results_list.append(this_row_results)
final_results_pd = pd.DataFrame(final_results_list,
index=result_crosstab_pd.index,
columns=result_crosstab_pd.columns)

sns.load_dataset(’iris’)
plt.figure()
sns.heatmap(final_results_pd,cmap="YlGn", annot=True, cbar=True)

**Step** **6:**
**Selectively building Model**
In this Step, based on the evaluation results in Step 5, the user can select high-capability APIs, for improving the difficult-to-train contexts in the built model, and produce an efficient process.
**Step 6-1: Preparing labeled image data**
Follow Step 1 to Step 3 in Section 3.1, the user collects and labels a moderate scale of image data required for building a model with machine learning (Generally, $n^{\prime }\geq 10n$). The user also selects the original image data with more prominent features for the previously known difficult-to-training contexts $c_{d}$ ($c_{d}\in C$), which may bring a buffer value to the accuracy of other contexts.
**Step 6-2: Selecting high-capability APIs to call**
As a addition in Step 4 in Section 3.1, the user first determine the difficult-to-train contexts $c_{d}$, and to select multiple high capability $API_{h}=\left\{api_{1},api_{2},\ldots ,api_{g}\right\}$ ($API_{h}\in API,g<p$) from evaluation results of the new Step 5-3. The selection approach as follows.(1) The user can select $API_{h}\left(g=1\right)$ that with maximum total scores of contexts, to ensure the built model with high overall accuracy.(2) The user can also select $API_{h}\left(g=2\right)$ with high complementarity of context evaluation results, to ensure the built model with high average accuracy.The approach (2) for home context sensing in most cases better than the (1), because of too much or too little will have an effect on the accuracy, efficiency, and complexity. Therefore, to select the small number of high-capability APIs rather than reduce the dimension of features can produce more applicable features related to $c_{i}$, for improving the process efficiency.Then, for every $c_{i}\in C$, $img_{ij}\in IMG\left(c_{i}\right)$, and $api_{k^{\prime }}\in API_{g}$, $api_{k^{\prime }}\left(img_{ij}\right)$ is invoked, and a set $Tag\left(img_{ij},api_{k^{\prime }}\right)=\left\{w_{1},w_{2},w_{3},\ldots \right\}$ of output tags is obtained. $Tag(img_{ij},api_{k^{\prime }})$ represents a recognition result for $api_{k^{\prime }}$ for an image $img_{ij}$ belonging to a context $c_{i}$. The size of $Tag(img_{ij},api_{k^{\prime }})$ varies for $img_{ij}$ and $api_{k^{\prime }}$. Since there are *m* contexts, $n^{\prime }$ images for each context, and *g* APIs, this step creates totally $m\times n^{\prime }\times g$ sets of output tags.
**Step 6-3: Encoding and combining features**
Follow Step 5-1 in Section 3.1, regarding every set $Tag(img_{ij},api_{k^{\prime }})$ of output tags as a document corpus, the user can first extract features from each document, by converting every set $Tag(img_{ij},api_{k^{\prime }})$ into a document vector $V\left(img_{ij},api_{k^{\prime }}\right)=\left\{v_{1},v_{2},\ldots \right\}$ (see Table 2). Then, for each $img_{ij}$, the user combines all $V(api_{k^{\prime }})$ into $⋃V(img_{ij},api_{k^{\prime }})$.Certainly, the another approach is to combine all $Tag(img_{ij},api_{k^{\prime }})$ into $⋃Tag(img_{ij},api_{k^{\prime }})$ in first, then converting them to document vectors. However, in this way, it could have influenced encoding results if there are the same tags output by $API_{h}$. We do not think that it means the features are highlighted in the related $img_{ij}$.
**Step 6-4: Building a model and Verifying results**
For each context $c_{i}$, the user split $⋃V(img_{ij},api_{k^{\prime }})$ into training data and test data. The user first applies a supervised machine learning algorithm *A* into the training data and corresponding labels for building a model *M*. The algorithm *A* include *Support Vector Machine (SVM)* [44], *Neural Network (NN)* [45], *Decision Tree* [46] and so on. Then, using the test data, the user evaluates the built model *M*, by checking the output $c_{i}$ with the labeled test($c_{i}$). The more *M* outputs the correct contexts, the *M* is more accurate.As the verification approach, the user can verify if the capability of the selected APIs the same as expected, by checking the accuracy of overall, average, context-wise and so on. The user can also check if the difficult-to-training contexts $c_{d}$ evaluated from the new Step 5-3 indeed difficult to recognize, and if the accuracy of them is improved.

## 4. Experimental Evaluation and Verification

This section introduces an experiment conducted for evaluating the capability of cognitive APIs, and verifying the evaluation results, to produce a flexible and efficient process of home context sensing.

### 4.1. Experimental Setup

In this experiment, we set the target space to be a smart home space, which is a part of our laboratory. For Step 1, we install a USB camera to acquire images of the daily activities of members of the laboratory. We develop a program that takes a snapshot with the USB camera every five seconds, and the images are cumulated in a server for nine months. In Step 2, we define seven contexts: “Dining together”, “General meeting”, “Nobody”, “One-to-one meeting”, “Personal study”, “Play games”, and “Room cleaning”. The representative images of each context and USB camera in this experiment is shown in Figure 3.

### 4.2. Evaluating Capability of Cognitive APIs

In Step 3, for each context, we selected 10 representative images considered to expose the context well. The selection is done by visual inspection so that the 10 images are chosen from different dates and times as possible. In Step 4, the images are sent to the five different APIs: Microsoft Azure Computer Vision (Azure) API [47], IBM Watson Visual Recognition (Watson) [16], Clarifai API [48], Imagga REST (Imagga) API [49], and ParallelDots API [50]. The total 350 sets of output tags (= 7 contexts × 10 images × 5 APIs) are obtained. The Step 5-1 to Step 5-3 have conducted in python by us (see Table 2, Table 3, and Table 4). In the new Step 5-1, we used Term Frequency - Inverse Document Frequency (TF-IDF) [34] to encode each set of output tags to a vector. In the new Step 5-2, we applied all document vectors into cosine similarity [21] and k-means++ [51] algorithms, producing a process of automatic classification. In the new Step 5-3, as the capability evaluation, for the clustering results of each API, we calculated the scores of each context with internal cohesion and external isolation concepts into the cross table. We also calculated the total score of each context and API for making a more detailed analysis in the table.

### 4.3. Building a Model to Verify

Based on the evaluation results of the new Step 5-3, we implemented to selectively building model. In Step 6-1, follow Step 1 to Step 3, we selected 100 representative images for each context. Especially, we selected the original images of the difficult-to-train context with more prominent features. In Step 6-2, we selected two APIs with most high-capability (i.e., Clarifai API and Imagga API), and respectively sent the representative images to them for obtaining the output tags. In Step 6-3, we respectively encoded the output tags to document vectors with the TF-IDF method, and combined all document vectors for each image. In Step 6-4, for each context, we split all document vectors into half, as training data and test data. We first applied the Multi-class Neural Network algorithm into the training data using Microsoft Azure Machine Learning [52]. Then, we using the test data to evaluating the recognition accuracy of the build model. We compared the capability of the selected high-capability APIs, and the accuracy of difficult-to-training contexts, with the evaluation results of the new Step 5-3.

### 4.4. Results

Figure 4 shows the capability evaluation results of five cognitive APIs in this experiment. Among the five evaluation results, the contexts that with relatively good stability include the results in “Play games” of Watson API, “Dining together” of Imagga API, and “Nobody” of Paralleldots. Because in the above results that other values are not in the row or column of the maximum value of that row. In contrast, the contexts that with relatively bad stability (i.e., difficult-to-training contexts) include the results in “Play games” of Clarifai API and ParallelDots API, “Room Cleaning” of Watson API, Clarifai API, and Imagga API. Table 1 shows the maximum value in each row of the capability evaluation results of each API from Figure 4. From the total score of each API, the APIs that with relatively good capability include Imagga API and Clarifai API. In contrast, the APIs that with relatively bad capability include Watson API and ParallelDots API. From the total score of each context, the contexts that easily to be recognized include “Nobody” and “General meeting”. In contrast, the contexts that difficult to be recognized include “Room cleaning” and “Play games”. Figure 5 shows the results by combining features of the selected APIs (Clarifai and Imagga APIs). From the main results of the metrics, the overall accuracy reached around 0.977, and the average accuracy reached around 0.993. From the results of the confusion matrix, the contexts that with accuracy reached 100% include “Dining together”, “General meeting”, “Nobody”, “One-to-one meeting”, and “Personal study”. The accuracy of “Play games” was 96.1%, and “Room Cleaning” was 88.2%. The result of the built model reached the same as we expected. Especially, it verified that the difficult-to-train contexts evaluated from the new Step 5-3 indeed difficult to recognize, but the accuracy of them to a large extent was improved.

### 4.5. Discussion

In this study, the factors that influence the home context sensing were many, which required to be considered. More specifically, the main factors in the different home contexts: (1) the position-change-degree of persons. (2) the number of persons existing. (3) the richness degree of objects existing. The contexts defined by us in this experiment covered the difference of the above (1) (2) (3) cases. In this way, we easily understand that the recognition accuracy of “Room cleaning” was not good in Figure 5, because of the (1) (2). A snapshot only represents the contents in a moment, which cannot retrieve more informative features on the time series. For another difficult home context “Play games”, the difference was existing among the labeled data due to the above (2). In the play games every time, the number of persons with the difference between two to five. Further more, in the results of Figure 4, we regard the maximum value of each row as the easy degree of that context to be recognized. We have put them into Table 1 for easier to check. However, as Figure 4 shows, the other values in the row or column of the maximum value of each row still inevitable in most situations. They reflected the disturbance items for each context were existing, which should be also considered in the final evaluation calculation. Such as letting the maximum values subtract the other values that in the same row and columns, it might a good way.

## 5. Conclusions

In this paper, a method that uses a small scale of labeled data to evaluate the capability of image-based cognitive APIs in advance, towards the flexible and efficient home context sensing, is proposed. From experimental evaluation and verification, the high-capability APIs and difficult-to-train contexts well within the response range of evaluated results, confirming the advantage that the predictability and efficiency of feature extraction with cognitive APIs are improved by the proposed method.

The initial thinking of our study is to realize a system, where a simple edge system just capturing, and pre-processing images are deployed at home. All heavy tasks of image recognition are delegated to the cognitive service in the cloud. However, in the complex and different environments by one household to another household, for the APIs with many difficult-to-training data, it is still a big challenge that, how to rapidly and accurately obtain the more fine-grained evaluation results in advance. We have tried some experiments to extract more valuable features from the return results of each API, such as the score or confidence value of each output tag. However, the way has not been found to utilize them, due to there existing the big difference in the range and distribution of that score values by the different APIs. As future work, we will try to find the edge computing techniques with image recognition for implementing a more smart home context sensing.

## Figures and Tables

**Figure 1 sensors-20-01442-f001:**
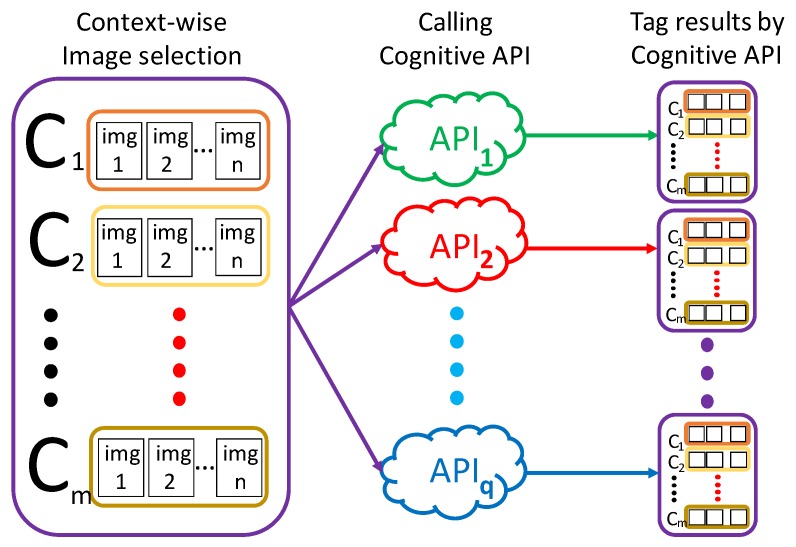
The flow from context label setting to analysis of results.

**Figure 2 sensors-20-01442-f002:**
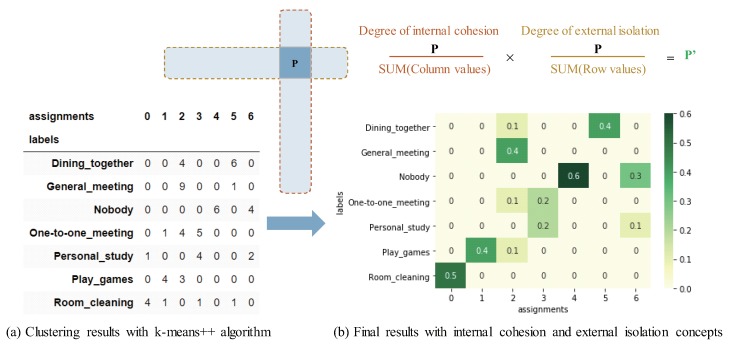
Example of the process for evaluating the capability of cognitive Application Program Interfaces (APIs).

**Figure 3 sensors-20-01442-f003:**
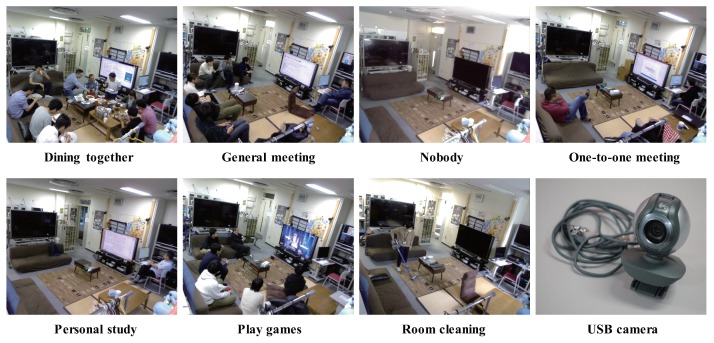
The representative images of each context and USB camera in this experiment.

**Figure 4 sensors-20-01442-f004:**
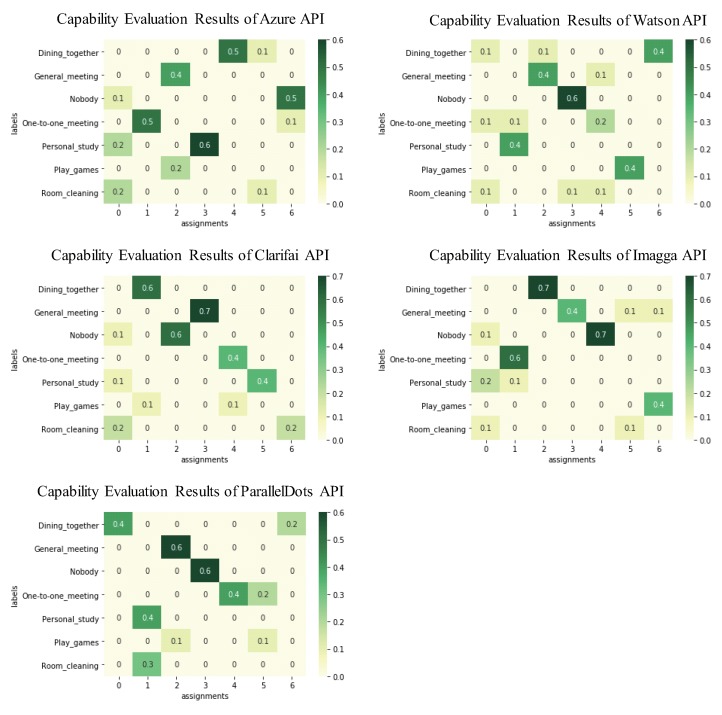
The capability evaluation results of five cognitive APIs in this experiment.

**Figure 5 sensors-20-01442-f005:**
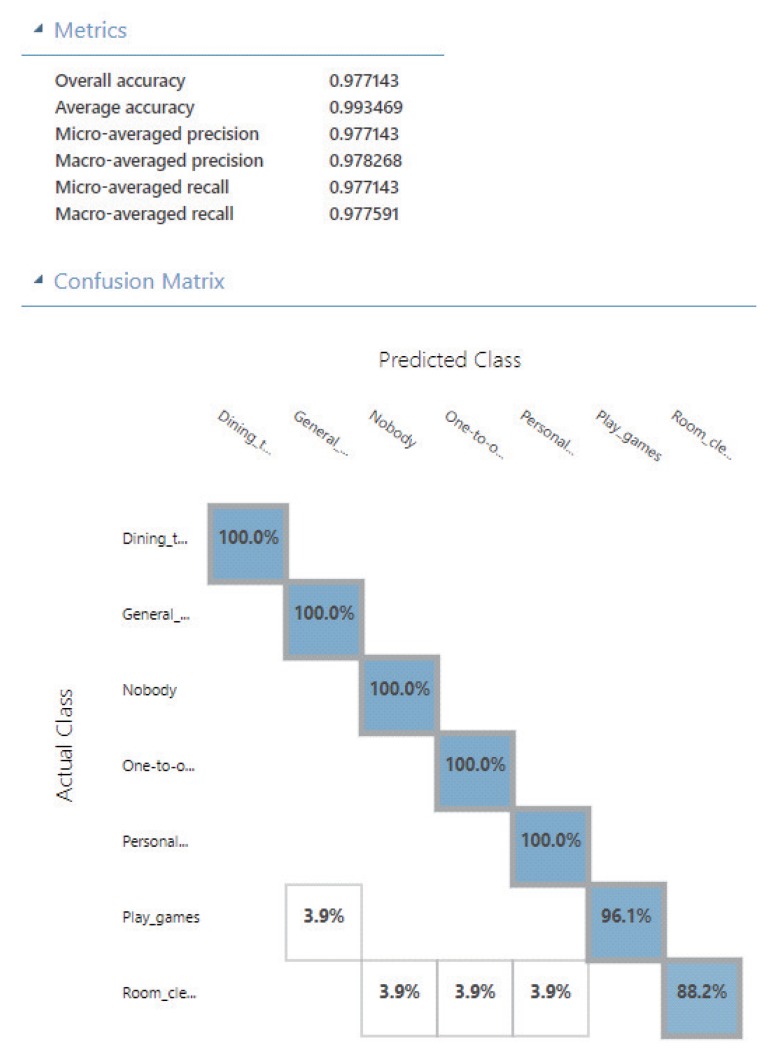
The results by combining features of the selected APIs (Clarifai and Imagga).

**Table 1 sensors-20-01442-t001:** The maximum value in each row of the capability evaluation results of each API from Figure 4.

Context Names	Azure API	Watson API	Clarifai API	Imagga API	ParallelDots API	Total
Dining together	0.5	0.4	0.6	0.7	0.4	2.6
General meeting	0.4	0.4	0.7	0.4	0.6	2.5
Nobody	0.5	0.6	0.6	0.7	0.6	3
One-to-one meeting	0.5	0.2	0.4	0.6	0.4	2.1
Personal study	0.6	0.4	0.4	0.2	0.4	2
Play games	0.2	0.4	0.1	0.4	0.1	1.2
Room cleaning	0.2	0.1	0.2	0.1	0.3	0.9
Total	2.9	2.5	3	3.1	2.8

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
