# Peer review of "Toward Flexible and Efficient Home Context Sensing: Capability Evaluation and Verification of Image-Based Cognitive APIs†"

_sensors, 2020, doi:10.3390/s20051442_

Round 1

Reviewer 1 Report

This paper presents a method that evaluates the feasibility of image-based cognitive APIs.

1) Similarity and contribution compared with original article, and recommendation.
The original research paper may be "Evaluating Feasibility of Image-Based Cognitive APIs for Home Context Sensing" which was presented in 2018 ICSPIS conference.
I'm not sure it is ok if the authors do not refer to the original paper, and please check the similarity of important research results. What are the added contributions when comparing with the conference paper?

2) In the evaluation part, readers may wonder about false alarm rate, mis-detection rate, and accuracy of the proposed scheme in realistic usage scenarios.

3) How about speed and complexity of the proposed scheme when compared with conventional method?

Reviewer 2 Report

I think this is an extension to the proceedings of 2018 International Conference on Signal Processing and Information Security (ICSPIS) with the title: "Evaluating Feasibility of Image-Based Cognitive APIs for Home Context Sensing".

The conceptual part is as well described in the earlier publication hence, the novelty is very minimal in this submission.

Regarding the results section: The pictorial representations were changed with respect to the previous publication.

It would be better to address the following: As there are issues with the cloud management such as latency, bandwidth, etc.,  It would be better to experiment with Fog computing architectures for better utilization of the Cognitive APIs IN THE HOME CONTEXT SENSING

Reviewer 3 Report

The paper presents an interesting approach for context recognition based on images and API REST. The most interesting part is to use external API in order to obtain a set of tags for an image. Then, this set is transformed into a vector. Finally, the vector is used to compute similarity with another vector.

However, I have major concerns about the paper. Although the paper presents an interesting approach, the quality of the paper makes hard to appreciate and understand well the work. 

1 - The authors should improve the clarity of the paper. Indeed, it is hard to distinguish exactly what are the contributions of the authors.

2 - The authors should improve the clarity of the results. For example, Figure 5 shows a kind of confusion matrix, with some discussions. However, it is difficult to link comments with Figure 5. The assignment numbers should be defined in the paper. 

3 - As a recommendation, the authors could place the related work section as Section 2. I prefer to read what is existing first, and then read the proposed work in order to focus on contributions. 

4 - Line 202, it seems to miss a reference.

5 - Please check if there are scam journals in reference (here an exhaustive list of it: https://predatoryjournals.com/journals/).

6 - Finally, the use of the unsupervised algorithm is not clear. Indeed, as the paper presents the work, a supervised algorithm is a better solution to the problem.  

Round 2

Reviewer 1 Report

I recommend this paper for publication.

Reviewer 2 Report

The authors have responded to my previous review and incorporated the required changes as per the suggestions.

1) The English corrections are required as the text contain sentences such as 

"....However, as the describe earlier......". what does it mean "as the describe"

2) The contribution in this paper is the step :5 Analyzing output tags.

The steps 5-2 and 5-3 require the evaluation description.

Reviewer 3 Report

The authors respond to all comments. Also, they added new results. It is a good paper presented an interesting method. 
